# Modeling the Optimal Conditions for Improved Efficacy and Crosslink Depth of Photo-Initiated Polymerization

**DOI:** 10.3390/polym11020217

**Published:** 2019-01-27

**Authors:** Jui-Teng Lin, Hsia-Wei Liu, Kuo-Ti Chen, Da-Chuan Cheng

**Affiliations:** 1New Vision Inc., 10F, No. 55, Sect.3, Xinbei Blvd, Xinzhuang, New Taipei City 242, Taiwan; jtlin55@gmail.com; 2Department of Life Science, Fu Jen Catholic University, No. 510, Zhongzheng Rd., Xinzhuang, New Taipei City 242, Taiwan; 079336@gmail.com; 3Graduate Institute of Applied Science and Engineering, Fu Jen Catholic University, Xinzhuang, New Taipei City 242, Taiwan; tony022199@msn.com; 4Department of Biomedical Imaging and Radiological Science, China Medical University, Taichung 404, Taiwan

**Keywords:** polymerization modeling, kinetic, photoinitiator, optimal efficacy, crosslinking

## Abstract

Optimal conditions for maximum efficacy of photoinitiated polymerization are theoretically presented. Analytic formulas are shown for the crosslink time, crosslink depth, and efficacy function. The roles of photoinitiator (PI) concentration, diffusion depth, and light intensity on the polymerization spatial and temporal profiles are presented for both uniform and non-uniform cases. For the type I mechanism, higher intensity may accelerate the polymer action process, but it suffers a lower steady-state efficacy. This may be overcome by a controlled re-supply of PI concentration during the light exposure. In challenging the conventional Beer–Lambert law (BLL), a generalized, time-dependent BLL (a Lin-law) is derived. This study, for the first time, presents analytic formulas for curing depth and crosslink time without the assumption of thin-film or spatial average. Various optimal conditions are developed for maximum efficacy based on a numerically-fit A-factor. Experimental data are analyzed for the role of PI concentration and light intensity on the gelation (crosslink) time and efficacy.

## 1. Introduction

Photoinitiated polymerization and crosslinking provide advantageous means over thermal-initiated polymerization, including fast and controllable reaction rates, and spatial and temporal control over the formation of the material, without the need for high temperatures or harsh conditions [1]. Tissue-engineering using scaffold-based procedures for chemical modification of polymers has been reported to improve its mechanical properties by crosslinking or polymerization with UV or visible light to produce gels or high-molecular-weight polymers [2,3,4,5,6,7]. The progress of light-responsive smart nanomaterials was recently review by Yang et al. [8]. Various crosslinking methods have been developed to stabilize collagen in aqueous solutions ex vivo, including physical interactions, chemical reactions, or photochemical polymerization [2,3,4]. The advantages and limitations of photo crosslinking have been discussed by many researchers [9,10,11,12,13,14,15,16,17], specially for the thiol-click reactions which include Michael-addition [17] and the thiol-ene reaction [10,11,12,13,18,19,20], where light sources both in UV [9,12] and visible spectra [16,19,20] have been used to initiate polymerization and crosslinking. 

The kinetics of photoinitiated polymerization systems (PPS) have been studied by many researchers for uniform photoinitiator (PI) distribution and for oversimplified cases without photolysis absorption or constant light intensity [21,22,23,24,25,26]. Previous models of PPS assumed either a constant light intensity [20] (for thin polymers), or a conventional Beer–Lambert law (BLL) [23,24,25,26,27,28,29,30,31,32] for the light intensity. For more realistic systems, one should expect non-uniform PI distribution, and the UV light may still be absorbed by the photolysis product besides the absorption of the monomer. To improve the efficiency and spatial uniformity (in the depth direction), particularly in a thick system (>1.0 cm), we have presented the numerical results using a focused light [33] and two-beam approach [34] for the case of uniform PI distribution; and analytic and computer modeling for the non-uniform case [35].

Radical photoinitiation systems are broadly divided into two classes associated with their radical generation mechanism following photon absorption. Type I (cleavage type) photoinitiators dissociate into two radicals following photon absorption, while type II initiating systems, in an excited state after photon absorption, abstract a hydrogen atom from a second, coinitiator species [20,29]. Sequential network formation has also been achieved with many different types of polymerization methods, such as thiol–Michael/acrylate hybrid, epoxy/acrylate curable resins, thiol–acrylate/thiol–acetoacetate thermosets, and thiol–ene/epoxy-based polymers [20]. More recently, wavelength-selective systems were developed to achieve precise temporal and spatial control on polymerization for customizable, highly tunable polymer behavior in optical lithography, photodegradable materials, and drug delivery [21,22,29].

Besides the industrial applications, such as microlithography, there are many biomedical and clinical applications of photopolymerization including the traditional UV-curing of dental resins [1,2], and the most recent clinical uses for corneal collagen crosslinkink [36,37]. Photodynamic therapy (PDT) using light-initiated oxygen free radicals offers biomedical applications in dermatology, orthopedics (tissue engineering), ophthalmology, and cancer treatments in various parts of human body, including early stage (micro-invasive) lung cancer, lung tumors (endobronchial, mesothelioma), skin, brain, colorectal, and breast cancer, chronic skin diseases (oriasis, vitiligo), and oral cavity (anti-bacterial, curing) [1,2,7,8,9,10]. Ophthalmology applications include age-related macular degeneration (AMD), coagulation of retina and corneal collagen crosslinking (CXL). Corneal collagen crosslinking is a clinical procedure for the treatment of keratoconus and ulcers, in which riboflavin solution (as a photosensitizer) was applied to the cornea followed by a UV light (at 365 nm) resulting in the crosslink (or stiffness increase) of corneal stroma collagen [35,36,37].

In general, both type-I (non-oxygen-mediated) and type-II (oxygen-mediated) reactions can occur simultaneously, and the ratio between these processes depends on the types and the concentrations of PI, substrate and oxygen, and kinetic rates [36,37]. Zhu et al. [31] and Kim et al. [32] reported the kinetics and macroscopic modeling of PPS for anti-cancer, which, however, are limited to the type-I oxygen-mediated process. Lin reported the kinetic modeling for both type-I and type-II mechanisms for the CXL application [36,37,38,39], where the temporal and spatial profiles of PI concentration and the efficacy were also reported [35]. The treated corneas have a much smaller thickness (approximately 500 μm, or 0.05 cm) than a typical thick polymer system (approximately 1.0 cm). Therefore, the PPS depth-profile and optimal features in thick polymers require further studies. Accelerated corneal crosslinking has been clinically used for faster procedure (within 3 to 10 min) using higher light intensity of 9 to 45 mW/cm^2^, in replacing the conventional 3 mW/cm^2^ which took 30 min [35,38]. However, to our knowledge, no efforts have been done for fast PPS in optically thick polymers using a high light intensity and under a controlled PI concentration. 

For practical and/or medical purpose, the preferred parameters of PPS include: minimum dose (or fluence), fast procedure, minimum cell toxicity, minimum concentration, maximum and uniform reactive depth, and maximum efficacy. However, some of these parameters are competing factors, and therefore, optimal condition is required for best outcomes. Furthermore, environment conditions such as the internal and external amount of oxygen and PI concentration control are critical in determining the efficacy. 

In this study, we will investigate the roles of PI initial concentration and light intensity and fluence (dose) on type-I and type-II efficacy, for both uniform and non-uniform cases. For optimal efficacy, strategy via controlled PI concentration will be presented, where re-supply of PI concentration during the PPS is defined by a polymerization (crosslink) time which is inverse proportional to the light intensity [35]. The validation of Bunsen–Roscoe reciprocal law (BRL) in type-I and type-II mechanisms will be discussed by the non-linear feature of efficacy. In challenging the BLL, a generalized, time-dependent BLL (Lin-law) is derived. For the first time, we will present the analytic formulas for curing depth and crosslink time without the assumption of thin film or spatial-average. Various optimal conditions will be developed for maximum efficacy based on a numerically-fit A-factor. Finally, the reported measurements of the role of light intensity and the PI concentration [12,18,39,40] will be analyzed by our formulas. While most previous models are limited to numerical results, this study will provide analytic formulas for in-depth analysis and as the guideline for future laboratory or clinical studies. 

## 2. Methods and Modeling Systems

### 2.1. Photochemical Kinetics 

As previously reviewed by Lin [37] for corneal crosslinking, the photochemical kinetics have three pathways which are revised for a more general polymer system and are briefly summarized as follows. We will limit the kinetic to the simple case of radical-mediated mechanism, although a more complex, two-step thiol-Michael mechanism, involving anionic centers reactive intermediates may also occur [29]. 

As shown in Figure 1, in the type-I pathway, the excited PI triple-state (T_3_) can interact directly with the substrate (A); or with the ground state oxygen (O_2_) to generate a superoxide anion (O^−^), which further reacts with oxygen to produce reactive radical (O^−^). In comparison, in the type-II pathway, T_3_ interacts with the ground state oxygen (O_2_) to form a reactive singlet oxygen (O*). In general, both type-I and type-II reactions can occur simultaneously, and the ratio between these processes depends on the types and the concentrations of PI, substrate, and oxygen, the kinetic rates involved in the process [36,37]. 

These factors also influence the overall photopolymerization efficacy, particularly the PI triplet state quantum yield (q) and its concentration. Furthermore, the specific protocols and the methods of PI instillations prior to and during the photopolymerization also affect the short- and long-term outcomes. The overall photopolymerization efficacy is proportional to the time integration of the light intensity, I(z, t) and the PI and oxygen concentration, C(z, t), and [O_2_]. The efficacy reaches a saturated (steady) state when C(z, t) or O_2_ is depleted by the light, where higher intensity depletes C(z, t) and [O_2_] faster, and therefore reaches a lower steady-state efficacy [37,38].

Referring to the kinetic pathways shown by Figure 1 and Reference [37], a set of quasi steady-state macroscopic kinetic equations for the time (t) and spatial (z) profiles of PI ground-state, C(z, t), oxygen molecule [O_2_], and light intensity, I(z, t), are constructed [31,36,37]:(1)$$\frac{\mathrm{dC}\left(z,t\right)}{\mathrm{dt}}=-b\left[g+g\prime \right]C$$
(2)$$\frac{d\left[O_{2}\right]}{\mathrm{dt}}=-\mathrm{sbCG}+P\left(z,t\right)$$
(3)$$\frac{\mathrm{dI}\left(z,t\right)}{\mathrm{dz}}=-A\prime \left(z,t\right)I\left(z,t\right)$$
(4)$$A\prime \left(z,t\right)=2.3\left[\left(a\prime -b\prime \right)C\left(z,t\right)+{b\prime C}_{0}\left(z\right)\right]+Q$$
where b = aqI(z, t); a = 83.6wa’; w is the light wavelength (in cm); a’ and b’ are the molar extinction coefficient (in 1/mM/%) of the initiator and the photolysis product, respectively; Q is the absorption coefficient of the monomer and the polymer repeat unit. I(z, t) has a unit of mW/cm^2^. In Equation (4), we have included the light intensity in the polymer given by a time-dependent, generalized Beer–Lambert law [35]. We have also included in Equation (2) the oxygen source term P(z, t) = p(1 − [O_2_]/[O_0_]), with a rate constant p to count for the situation where there is an external continuing supply, or natural replenishment (at a rate of p), besides the initial oxygen in the polymer.

Equation (1) defines the type-I process given by, g = k_8_[A]G_0_/k_3_, G_0_ = 1/([O_2_] + k + K’); and type-II process given by, g’ = K’(C + c’)G(z), with G(z) = [O_2_]G_0_, k = (k_5_ + k_8_[A])/k_3_; K’ = k_12_/(k_6_ + k_12_(C + c’) + k_72_[A]); c’ is a low concentration correction related to the diffusion of singlet oxygen [30]. [A] is the substrate concentration; q is the triplet state quantum yield given by q = k_2_/(k_1_ + k_2_); s = s_1_ + s_2_, with s_1_ and s_2_ being the fraction of [O_2_] converted to other reactive radicals and the singlet oxygen, respectively, in type-I and type-II [36]. All other rate constants, k_j_, k_ij_ are defined by the reaction paths shown in Figure 1. Greater details of Equations (1) to (4) can be found in our previous articles cited as Reference [35] and [37]. 

We note that Equations (1) to (4) were also presented by Kim et al [31,32] for the anti-cancer kinetics. However, they have assumed a constant light intensity, i.e., A’(z, t) is a constant in Equation (4). They also ignored the contribution from the type-I term, g or k_8_[A], since type-II is dominant in their anti-cancer process. Most of the previous models [23,24,25,26,27,28] have also ignored the dynamic absorption factor, A’(z, t) shown by Equation (4), due to the strong depletion of PI concentration [35]. Greater detail of the kinetics shown by Figure 1 and the complex coupled equations prior to the use of quasi steady-state condition have been discussed in corneal crosslink for both type-I and type-II [36,37], and in thick polymer systems for type-I [34]. This article will focus on analytic formulas and various strategies for optimal efficacy.

### 2.2. Generalized Beer–Lambert Law

The light intensity is given by the integration of A’(z, t) of Equation (4), which, in general, is time- and z-dependent due to the depletion of C(z, t). The first-order solution of Equation (1) is given by C(z, t) = C_0_F(z)exp(−Bt), with B = bg = aqgI’(z), and F(z) is the initial PI concentration distribution, i.e., F(z) = C(z, t = 0), defined by a diffusion depth (D). I’(z,) is a time averaged light intensity, assuming b and g are time-independent. One may also use an averaged A’(z, t) in Equation (4), which has an initial value A_1_, (with b’ = 0) and a steady state value A_2_ (with C = 0), given by: A_1_’ = 2.3a’C_0_F’ + Q, and A_2_’ = 2.3b’C_0_F’ + Q, with F’(z) = 1 − 0.25z/D given by the integral of F(z); the mean value is given by A = 0.5(A_1_’ + A_2_’). In above simplified cases, the light intensity is time-independent and its z-dependence follows the BLL. However, depletion of C(z, t) will also affect the time-dependent profiles of the intensity, I(z, t), which in general, will not follow the BLL, and should be governed by a generalized, time-dependent BLL (the Lin-law) first presented by Lin et al [35,36]. Greater detail of Lin’s law is derived as follows. 

Using the Taylor expansion exp(−A’z) = 1 − A’z, the first-order C(z, t) = C_0_exp[−B’t(1−A’z)] and integrating A’(z, t) of Equation (4) over z, we obtain the Lin-law: I(z, t) = I_0_exp(−Az), with A(z, t) = 2.3b’C_0_ + Q + A_2_(t), and the time-dependent component, A_2_ = 2.3(a’ − b’)(1 + 0.5ztB”A’)]pC_0,_ with p = exp(−B”t), B” = aqgI_0_, A’ = 0.5(A_1_’ + A_2_’). One may further simplify, using p = 1 − tB”, to obtain A(z, t) = A_0_ – A_1_t, with A_0_ = 2.3a’C_0_ + Q, and A_1_ = 2.3(a’− b’)B”(1 − 0.5A’z)C_0_, which explicitly shows that A(z, t) of Lin-law is a decreasing function of time (t) and leads to the dynamic light intensity, being an increasing function of time due to the depletion of PI concentration, C(z, t), in Equation (4). These photobleaching effects have been reported for the normal case that a’ > b’; however, photodarkening effects were also reported when a’ < b’ [26]. The Lin-law for A(z, t) covers both the initial value (A_1_’, at t = 0) and the steady state value (A_2_’, at t >> 1/B’) defined earlier. Compared to BLL, the Lin-law has two modifications via A_2_(z, t): the time dependent term p = exp(−tB’), and the z-dependent term, 0.5ztB’A’, counting for the feature that C(z, t) is an increasing function of z (for t > 0), i.e., the crosslink starts from the surface layer. Therefore, BLL can be considered as a special case of Lin-law for this specific application field. 

Photobleach effects of PI may be ignored only for systems having an optically thin film, or when the light dose is small, i.e., when tB’ << 1 and PI concentration is not depleted. The Lin-law provides a more accurate analytic formula for both I(z, t) and C(z, t), compared to the exact numerical solution, than the time-average law of A’(z, t), which is time-independent. Comparing analytic formulas using a fit A-factor and the exact numerical solution of Equations (1) to (4) will be shown later.

### 2.3. Efficacy Profiles 

Photopolymerization efficacy (Ef) is defined by the percentage of amount of monomers converted to polymers, i.e., E_f_ =1 − [M]/[M_0_], where [M] and [M_0_] are the converted and unconverted monomer concentration, respecyively. It may be further defined by an S function by E_f_ = 1 − exp(−S) [26,36]. For type-I S-function, we will replace g = k_8_[A]G_0_/k_3_, by an overall rate constant (K) including all polymerization chain reactions. Using a fit A-factor given by A(t)=0.5 × 2.3(a’ + b’)exp(−mt)C_0_ + Q, for A’(z, t) in Equation (4), with m being a fit parameter to the exact numerical solution of Equations (1) to (4). The analytic S-function (for type-I) given by the time integral of [aqgKC(z, t)I(z, t)]^0.5^ is obtained as follows [36]:(5)$$S=\sqrt{2{\mathrm{KC}}_{0}/B\prime }\;E\prime$$
(6)E′=[1−exp(−B′t)]
where B’ = aqgI_0_X’(z), and a fit X’(z, t) = exp(−Az).

Equation (5) reduces to our previous formula [35] for the special case of fit parameter m = 0, or B’ = B using a time averaged light intensity. In general, the fit m = 0.001 to 0.002, depending on the values of a’, b’, and C_0_; a numerical example will be shown later. Notably, Equation (5) is a non-linear function of the light dose (E_0_) given by the Taylor expansion of its transient term (for B’t < 1) E’= 0.5b”E_0_(1 − B’t), and S = [0.5KB’C_0_]^0.5^(1 − 0.5B’t), with b” = aqgX’(z), which does not follow the BRL in the above type-I process. In contrast, type-II efficacy, given by the time integral of I(z, t)C(z, t)[O_2_] follows the BRL [37]. We note that the steady-state (with E’ = 1) S function is proportional to [C_0_F(z) exp(zA_0_)]^0.5^, which is an increasing function of C_0_ and z. Numerical results of Equation (1) compared with Equation (5) will be shown later. 

The significance of Equation (5) with the fit A-factor is that it provides the simplest formula to find various optimal conditions, analytically, without involving complex simulations based on the exact solution of coupled Equation (1), which is both temporally and spatially dependent. We note that Equation (1) can be analytically solved only for the special case of b’ = Q = 0, or an assumption of spatial average over the PI concentration, C(z, t) [29]. 

### 2.4. Optimal Efficacy 

S has a maximum value at a depth (z = z*) given by taking dS/dz = 0. Using the analytic formula of Equation (5), for the case of F = 1.0 (or uniform PI concentration, with D >> 1.0 cm), and using an averaged A = 0.5 × 2.3(a’ + b’)C_0_ + Q, z* is given by the condition of B’t = 1.25, which leads to an analytic formula z* = (1/A) ln(aqgE_0_/1.25), and the corresponding PI concentration is given by C_0_* = ([ln(a”E_0_/1.25)]/z − 2.3Q)/[1.13(a’ + b’)]; the maximum S is given by E’ = 0.714. For the case of non-uniform PI concentration (with D is 0.5 to 1.0 cm), these optimal conditions require numerical calculations to find z*. Similarly, S also has an optimal intensity, I*, given by taking dS/dI = 0. We obtain I* = [2/(aqgt)]exp(Az), which is inverse proportional to t. The above optimal features will be shown later by numerical value of S. 

### 2.5. Depletion Time (T*) 

For a PI initial distribution function given by C_0_(z) = C_0_F(z), with F(z) = 1 − 0.5z/D, and D be the distribution depth, the solution of Equation (1c) gives I_0_(z, 0) = I_0_(1 − 0.25z/D). When D ≫ 1.0 cm, F = 1 represents a flat (or uniform) PI distribution. Analytic solution of Equation (1) is needed to derive the formulas of depletion time (T*). For g ≫ g’, we obtain a first-order solution, C(z, t) = C_0_exp(−Bt). A depletion time T* may be defined by when C(z, t) is depleted to e^−M^ of its initial value, with M is an integral. M = 1 has been chosen conventionally to define a e^−1^ (0.366) decaying rate. In this study, we will choose M = 4 to describe an almost completed depletion, i.e., C(z, t) is depleted to e^−4^ = 0.018 of its initial value. We obtain an analytic formula T*(z) = T_0_exp(Az), where T_0_ is the surface depletion time given by T_0_ = M/(aqgI_0_), which is inversely proportional to the light initial intensity. We note that T* is related to the saturation time (T_S_) defined by Equation (2b) when E’ = 0.87, or B’T_S_ = 2, which gives (when higher order terms of B’ are neglected) T_S_ = 2/B’ = [4/(aqgI_0_)] exp(Az) = T_0_ exp(Az), with the surface saturation time given by T_0_ = 4/(aqgI_0_). We note that the saturation time (T_S_) equals the depletion time (T*), when M = 4, and it is also proportional to the crosslink time, to be discussed later. Our formulas for T* and T_S_ can be calculated by the measured PI concentration temporal profile, which defines the rate constant (aqg), at a given light intensity (I_0_).

### 2.6. Volume Efficacy and Crosslink Time and Depth

Curing (crosslink) depth (z_S_), crosslink time (T_C_), and induction time (T_D_) have been calculated by Lee et al. [28], Cabral et al. [27], Hatanaka [29], and Lin et al. [34,35] based on various definitions. Lee et al. [28] defined the critical threshold for gelation by a critical conversion at which point a gelation is formed. Cabral et al. [27] presented numerical results for the general case, but their analytic formulas, Equations (5) to (8), are only for the thin-film cases in which the PI bleaching was ignored. They defined an “induction time” (T_D_) given by a measured threshold efficacy of 5% to 0.95%, depending on types of materials. The associate moving front (solid/liquid interface) was defined as a cure depth (z_C_), shown by their Equation (7). Cabral et al. [27] also defined a “saturation time” (T_S_) given by their Equation (8) sat as the time required for full (99%) conversion of the resist layer of a given thickness. In comparison, this study defines T_S_ based on Equation (5), when S reaches 87% (or 1−e^−1^) of its steady-state. 

A similar definition for the curing depth (z_C_) was reported by Lee at al. [28], and Hatanaka [29]. However, Lee et al. assumed a BLL for the light intensity, therefore their calculated cure depth (z_C_) was underestimated, without including the PI bleaching in A(z, t). Hatanaka [29] reported calculations including the PI bleaching effects, but he assumed a spatially averaged PI concentration, and only numerical results were presented. This study, for the first time, presents the analytic formulas for curing depth (z_C_) and crosslink time (T_C_) without the assumption of thin film or spatial-average. The averaged A-factor given by A = 0.5 × 2.3(a’ + b’)C_0_ + Q will be used to derive z_C_ and T_C_. 

The critical crosslink depth defined by when the transient value (for B’t < 1) of S in Equation (6) is larger than a threshold value (d). Using the transient value of E’ and solving for z = z_C_ for S = d, we obtain the crosslink depth, up to the second-order approximation,
(7)$$z_{C}=\left(1/A\right)\mathrm{lnY}$$
(8)$$Y=Y_{0}/\left[1-0.5b\prime \prime E_{0}/Y_{0}\right]$$
(9)$$Y_{0}=b\prime \prime {\mathrm{KtE}}_{0}C_{0}/d^{2}$$
With b” = aqgX(z), X(z) = exp(−Az). The critical crosslink depth defines the curing point such that gelation occurs (in the transient state) for z < z_C_ (with S>d), and no gelation for z > z_C_ (with S < d). We note that z_C_ has an optimal value of C_0_ given by dz_C_/dC_0_ = 0, for small Bt < 1, whereas it is a decreasing function of C_0_ for Bt ≫ 1 (or E’ = 1). These features were also reported by the measured data of Lee et al. [27] (referring to their Figure 4). However, their formula, assuming a constant PI concentration for z_C,_ is not as accurate as our Equation (3). Equation (17) of Lee et al. [28] is our special case when Y = Y_0_ in our Equation (3b), or the non-linear term, 0.5aqgX(z)E_0_/Y_0_, is neglected; and when A = 2.3a’C_0_ + Q, with Q = 0. Similarly, Equations (7) and (8) of Cabral et al. [27], also based on a thin-film assumption as that of Lee et al. We note that the pattern height (h) defined by Equation (7) of Cabral et al. is closely related to our curing depth (z_C_). However, the conversion efficacy of Cabral et al., shown by their Equation (1), is for a type-II process, whereas the S-function given by Equation (5) of this study and by Equation (15) of Hatanaka [28] are for a type-I process. We note that the crosslink depth has two values when Bt > d’, with d’ defined by when dS/dz > 0, whereas dS/dz < 0 for Bt < 1. This reverse feature will be further analyzed by the numerical results of Equation (5) to be shown later.

The crosslink time (T_C_) (at any depth z) may be derived by solving Equation (7) for t = T_C_, and z_C_ = z, We obtain
(10)$$T_{C}=T_{0}{X\prime }^{0.5}/[1-0.5b\prime \prime I_{0}/(T_{0}X^{0.5})]$$

With the surface crosslink time T_0_ = d/(b”KC_0_I_0_)^0.5^, noting that T_C_ is an exponentially increasing function of the depth (z). The crosslink time provides the gelation time needed to achieve a gelation depth of z with monomer-polymer conversion efficacy (Ef) higher than a threshold value of 1−exp(−d). For example, for threshold value d = 2.0, Ef = 1− exp(−d) = 0.87. We note that the “saturation time” defined by Equation (8) of Cabral et al. [26] is closely related to our crosslink time (T_C_), except that the threshold conversion E_f_ = 0.87 (this work), and 0.99 (Cabral et al.); noting that S = ln [1/(1 − E_f_)]. Cabral et al. also defined an “induction time” by a low threshold value of E_f_ = 0.1% to 0.5%, for the minimum time required for the surface layer to first become a solid. Therefore, the “induction time” is related to our T_0_ (surface crosslink time).

One may also further define a spatially averaged volume efficacy given by
(11)$$V=\left(1/V_{0}\right)\int_{0}^{\mathrm{zc}}S\left(z,t\right)\mathrm{dz}$$
(12)$$V_{0}=\int_{0}^{L}S\left(z,t\right)\mathrm{dz}$$
where L is the total polymer thickness. Given the transient value of S, we obtain V = (1 − X’^0.5^)/(1 − X^0.5^), with X’ = exp(−z_C_A_0_) and X = exp(−LA_0_). For small zA_0_ < 1 and LA_0_ < 1, V’=(z’/L)(1 − 0.5z’A_0_), which is a more accurate formula for the volume efficacy defined earlier as V = z_C_/L, based on a constant S value [35,39]. Greater detail and numerical exact solution of z_C_, T_C_, and V will be published elsewhere.

## 3. Results and Discussions

The Abbreviations, key parameters and formulas of this study are summarized in Table 1. 

### 3.1. Concentration Profiles

Numerical results of Equation (1) are shown in Figure 2 for the PI concentration dynamic profiles. One may see that depletion of PI starts from the polymer surface, and gradually into the volume (z > 0). We note that the PI concentration profile is an increasing function of z for the uniform case. In contrast, the non-uniform case shows a decreasing function of z. 

### 3.2. Efficacy Profiles 

Depending on the types of biomaterials and PI, a wide range of the PI absorption constant, concentration, and rate constants have been reported [12,25,27,28]. We chose typical values of the related parameters such that the crosslink time was in the range of 50 to 200 s, crosslink depth of 0.5 to 1.5 cm, PI concentration of 2 to 10 mM, and light intensity of 5 to 30 mW/cm^2^. These parameters lead to an S function value of 0.5 to 4, or efficacy, E_f_ = 1−exp(−S), of 40% to 98%, and covers both the transient and saturated states. Furthermore, the PI concentration cannot be too high to cause cell death. For example, photo-encapsulation of cells with 10 mW/cm^2^ exposure at 405 nm with 2.2 mM LAP is achieved in 5 min and results in 96% cell viability as reported by Fairbank et al. [12].

We will investigate the roles of C_0_, I_0_ and D on the spatial (z) and temporal (t) profiles of S, for both uniform and non-uniform cases by the numerical solutions of Equation (1), with results analyzed by our analytic formulas. All figures shown in the following are numerical solutions of Equation (1), except one of the figures which is based on Equation (5) and a fit A-factor with m = 0.001.

Figure 3 shows the time profiles of S versus time (left figure) and versus dose (right figure) for various light intensities I_0_ = (10, 15, 20, 30) mW/cm^2^, for D = 1.0 cm, C_0_ = 3 mM, at z = 0.5 cm. Figure 4 shows the spatial profiles of S for variou s exposure time, t = (100, 200, 300, 400) s, and compares profiles for low and high light intensity at I_0_ = 10 (left figure) and 20 (right figure) mW/cm^2^, for the non-uniform case with D = 1.0 cm. Figure 5 shows the effects of distribution depth (D), for a fixed light intensity.

Figure 6 shows S versus PI concentration (C_0_), for z = 0 and 0.5 cm, for D = 1.0 cm, I_0_ = 10 mW/cm^2^. Figure 7 shows profiles of S versus light intensity (I_0_), for z = 0 and 0.5 cm, for =pD = 1.0 cm, C_0_ = 2 mM, for t = 100 s (red curve) and 200 s (green curve). Figure 8 shows the spatial profiles of S based on the analytic formula of Equation (5), with fit parameter m = 0.001, for various PI concentrations at C_0_ = 2.0 and 3.0 mM, and various exposure times (for the case of uniform distribution, with D ≫ 10 cm). Figure 9 shows numerical results for S versus PI concentration at various times, t = (100, 200, 400) s, at thickness z = 1.5 cm, and for D ≫ 1.0 cm (F = 1).

We note that the transient factor E’ is scaled by (aqg) and the S function is scaled by S_0_ = [4K/(aqg)]^0.5^. Therefore, the above profiles may be easily reproduced for a given value of K and aqg, when PI with different absorption coefficient (a’), quantum yield (q) or effective kinetic rate constant (K) are used. 

### 3.3. Discussion of Numerical Results

From the above figures, and Equation (5), we are able to summarize the important “common” features for type-I PPS. These common features should cover both thin and thick polymers, including corneal crosslinking and polymer gelation. Unlike the assumption of a constant light intensity, which is valid only for very thin polymer, our formulas (including the PI depletion and dynamic of light intensity) are valid for all polymer thickness. 

(i) Figure 3 (left figure) demonstrates that higher light intensity has a faster rising efficacy, but a lower steady-state value due to its faster depletion of PI concentration. It is also shown by our formula, Equation (5), that the efficacy at transient state (for small dose) is proportional to tI_0_^0.5^. This new finding is similar to the crosslink depth (z_C_) which is an increasing function of I_0_t^2^, rather than light dose E_0_ = I_0_t. However, at steady-state (with E’ = 1), it is a non-linear function of [C_0_/I_0_]^0.5^ or [t/E_0_]^0.5^. This non-linear scaling law predicts the clinical data more accurate than the linear theory of Bunsen Roscoe law (BRL) [35,37].

(ii) Figure 3 (right figure) demonstrates that for the same dose (E_0_), all light intensity has the same crosslink time, as shown by Equation (6), E’ = 1 − exp(−B’t), with B’t = bg = aqgE_0_ which depends only on E_0_, rather than I_0_. 

(iii) As shown by Figure 4, higher light intensity has smaller steady-state efficacy, reduced by a factor of 1.43 when the intensity is doubled (for the same dose).

(iv) As shown by Figure 5, small diffusion depth (with D = 1.0 cm) has a more uniform but lower S profile; whereas large D (or flat, uniform PI concentration) has higher efficacy, but non-uniform profile. 

(v) As shown by Figure 6, optimal PI concentration (for maximum S) exists for z > 0, but not for z = 0. This optimal feature only exists in thick polymers (with z > 0.5 cm) and very high PI concentration, C_0_ > 30 mM. Under normal situation (with C_0_ < 10 mM), this optimal value does not exist. 

(vi) Similarly, as shown by Figure 7, there is an optimal light intensity (I*) for a given time, but not for light dose (shown by Figure 3). Moreover, as predicted by Equation (5), higher intensity has smaller efficacy (at steady-state, E’ = 1). To overcome this drawback of high light intensity, a novel method will be discussed later. These results are also predicted by our analytic formula, I* = [2/(aqgt)] exp(Az), which is inverse proportional to t. Furthermore, at steady state (with Bt ≫ 1), S has no optimal value and it is proportional to I_0_^−0.5^, as also predict by Equation (5).

(vii) As shown by Figure 8, based on Equation (5), we found several significant new features: (a) higher PI concentration has high S, or efficacy; (b) S is a decreasing function of z for t < t’, whereas it becomes a convex curves for t > t’, where the transition point t’ is defined by when dS/dz = 0, with t’ = 70 s (as shown by Figure 8); (c) for t > t’, z_C_ has two values of 1.0 and 1.5 cm (for C_0_ = 2.0 mM); and 0.2 and 1.5 cm (for C_0_ = 3.0 mM), as shown by red dots; (d) the red curve of Figure 8A is below the threshold value (S’ = 2.0), whereas Figure 8B, with a higher C_0_ = 3.0 mM, shown by a red curve that reaches S’ at t = 165 s, which is defined as the crosslink time (T_C_). Figure 8 demonstrates that high exposure time (or light dose) and higher PI concentration offer a higher S value (and efficacy) and wider range of crosslink depth, which also leads to a larger volume efficacy to be discussed later. 

(viii) As shown by the numerical results of Figure 9, the optimal concentration is proportional to the light dose as predicted by our formula: C_0_* = ([ln(a”E_0_/1.25)]/z − 2.3Q)/(2A), derived from analytic formula, Equation (5). Figure 9 also shows the height of the maximum S. These optimal conditions will require a lot of simulation times to solve Equation (1). In contrast, the analytic formula of Equation (5) can easily produce these curves in minimal time, and also predict the maximum height of S analytically.

### 3.4. The Role of Oxygen and Competing Process

As shown by Equation (1), type-I and type-II processes are governed, respectively, by g and g’. In the transient stage, both type-I and type-II coexist until the oxygen is completely depleted; then type-I dominates before the oxygen is resupplied or replenished. In general, the ratio between these two competing processes, or g/g’, depends on the types and initial concentrations of PI, concentrations of substrate and oxygen, and the kinetic rates involved in the process [36,37]. For type-I dominant crosslink, it is well known that oxygen inhibits free-radical polymerization, thereby reducing type-I crosslink efficiency [38,39,40]. This feature may be realized mathematically by Equation (5), with g = k_8_[A]/([O_2_] + k + K’) which is a decreasing function of [O_2_]. In contrast, type-II dominant procedure, such as photo-induced anti-cancer, governed by g’ = K’(C + c’)G_0_[O_2_], in which oxygen is required to produce enough singlet oxygen which kills the cancer cells. Therefore, our proposed strategy based on the resupply of oxygen and PI concentration would enhance the efficacy of type-II and type-I, respective, but not both. 

### 3.5. Analysis of Experimental Results

The reported measurements of Fairbank et al. [12], Shih et al. [19], O’Brart et al. [40], and Holmes et al. [41] for the crossover time, gelation kinetic profile and role of PI concentration are analyzed by our formulas as follows. 

Figure 3b of Fairbank et al. [12] indicated that the crossover (crosslink) time is a decreasing function of light intensity and the absorption constant (a‘). Our crosslink time formula, on surface (z = 0), T_0_ (in seconds) = 4/(aqgI_0_) = 1000/I_0_, for aqg = 0.004. Therefore, T_0_ = 100 s. For I_0_ = 10 mW/cm^2^, and A = 0.46 (1/cm), at z = 1.0 cm, T* = 100 exp(Az) = 158 s, which is comparable to that of Fairbank et al. for L = 0.5 mW/cm^3^ in lithium phenyl-2,4,6-trimethylbenzoylphosphinate (LAP) curve. The measured crossover time shows the same trend as our theory that crosslink time is a decreasing function of light intensity. Moreover, Fairbank et al. [12] also showed the time required to reach the gel point during the solution polymerization of poly(ethylene glycol) diacrylate (PEGDA) is lower for LAP than for I2959 (at 365 nm) at comparable intensities and initiator concentrations. This may be realized by our formula that Tc is predicted to be inverse proportional to (a’), the molar extinction coefficient, which is 0.218 (1/mM/cm), and 0.04 (1/mM/cm), for LAP and I2959, respectively [12]. Therefore, our formula predicts the S-function of LAP is approximately 5 times of I2959.

Our formula, Equation (5), also predicts that the steady-state S is proportional to the square-root of the PI concentration (C_0_). Therefore, the crosslink efficacy, defined by E_f_ = 1−exp(−S), is also an increasing function of C_0_. This feature has been clinically reported by O’Brart et al. in corneal crosslinking [40], using riboflavin solution as the PI and initiated by a UVA light at 365 nm. The role of PI concentration shown by Figure 3a of Fairbank et al. [12] is also predicted by our formula, Equation (5) that higher PI concentration offers higher crosslink efficacy and less gelation time. The optimal role of PI concentration is shown in Figure 9 of this study. 

The role of PI concentration was also shown by Table 1 of Holmes et al. [41], where (for LAP) increasing the PI concentration from 0.1% to 0.5% (*w*/*v*) in the thiol-ene mixture resulted in a 15-fold increase in the storage modulus. This increasing feature may be analyzed by our S-function, Equation (5), given by a steady-state formula $S=\sqrt{(C_{0}I_{0})\exp\left(\mathrm{Az}\right)}$. For C_0_ increases by five times, and for Az = 0.9, we calculate S-function increases by a factor of 2.23 × exp(2Az) = 14.5, which is comparable to the increase of storage modulus; noting that when C_0_ is 5 times, A is also five times, given by A = 2.3*b*’C_0_F(z) + *Q*, if *Q* ≪ 0.1 (1/cm). Figure 3 of Holems et al. [40] shows the gelation kinetic profiles which have a strong similarity to our Figure 3 based on Equation (5). The storage modulus was found to increase with time and UV exposure until a plateau was reached within 300 s, indicating no further elastic properties (complete chemical gel). The plateau-time corresponds to our crosslink time (T’) defined earlier. Similarly, the measured data of Shih et al. [19] (in their Figure 1) showed that crosslinking of thiol-norbornene PEG-peptide hydrogels in visible light was very similar to our Figure 3, except the time scale which depends on the types of PI and light used in the process. Unfortunately, Shih et al. [19] and Holems et al. [41] did not measure the profiles for different light intensity, as shown in our Figure 3, to justify our predicted feature that higher light intensity is less efficient in gelation. However, our predicted feature has been clinically demonstrated in ophthalmic system for CXL [37,39].

The above examples demonstrate that our formulas predict very well the measured results, at least the overall trends. However, the accuracy of our formulas will require accurate measurement of the parameters involved, such as the rate constant (K), the quantum yield (q), the molar extinction coefficient of the initiator (a’), the photolysis product (b’), and the monomer and the polymer repeat unit (Q) et al. In addition, further experimental measurements should also include the roles of PI concentration and light intensity.

## 4. Discussions for Optimal Strategy 

The goal of an optimized photo-click hydrogel system is to identify key influencing factors to enable fast gelation (<2 min), minimal photoinitiator-induced toxic response, maximum crosslink depth and uniformity, maximum efficacy, and tunable hydrogel elasticity. In the following, we will discuss the above issues with our analytic formulas and the numerically produced figures, including: (i) crosslink depth and uniformity, (ii) optimized ratio of PI concentration (C_0_) and light intensity (I_0_), (iii) strategy for improved efficacy, (d) the role of oxygen and competing process.

### 4.1. Crosslink Depth and Uniformity

The crosslink depth (z’) formula shows that higher Rf concentration results in an increased (or larger S_1_) but more superficial (or small z’) cross-linking effect, as also clinically indicated by O’Brart et al. [39]. For a given C_0_, deeper crosslink depth may be achieved by larger light dose (or fluence), E_0_. However, to achieve clinically acceptable crosslink efficacy by a minimal E_0_, one requires an optimal range of C_0_. such that z’ = 0.8 to 1.2 cm, and S = 1.5 to 2.0, or efficacy E_f_ = 1 − exp(−S) = 0.78 to 0.86.

As shown by Figure 5, small diffusion depth (with D = 1.0 cm) has a more uniform but lower S profile; whereas large D (or flat, uniform PI concentration) has higher efficacy, but non-uniform profile. Moreover, as shown by Figure 6, higher PI concentration (with C_0_ = 3.0 mM) has higher efficacy but a more non-uniform profile, compared to Figure 3 (with C_0_ = 2.0 mM). Therefore, the optimal design is to have D = 0.4 to 0.6 cm; and C_0_ = 1.5 to 2.5 mM. 

### 4.2. Optimized (C_0_/I_0_) Ratio 

As shown by our type-I S-formula, Equation (5) higher PI concentration (C_0_) and lower light intensity (I_0_) provides higher efficacy. Higher PI concentration provides higher efficacy, but also suffers more toxicity to the cells. Higher light intensity (I_0_) may accelerate the crosslink, but has lower efficacy (for the same dose). Therefore, optimal ratio of C_0_/I_0_ is required for accelerated and safe crosslink. Furthermore, there is an optimal range for C_0_ to make the S function larger than the threshed value (d) for a specified crosslink depth (referred to Figure 9). 

### 4.3. Transition Feature of S Function

As shown by Equation (5) and Figure 8, the transient profile (for Bt < 1) of S function is proportional to exp(−Az), an exponentially decreasing function of z. However, for long exposure time, Bt > 1, the z-dependence is reversed and S becomes an increasing function of z (for small z) and decreasing function of z (for large z). This new feature results from the competition between light intensity and PI concentration, where the monomer-polymer conversion rate has an optimal z. As shown in Figure 7; Figure 8, there is an optimal light intensity (or dose); the optimal I* is the transition point, where the dependence of S over B = aqg I_0_exp(−A’z) changes from a positive correlation to negative, i.e., a reverse feature.

### 4.4. Accelerated Photopolymerization

Another design for optimized optimized photopolymerization of hydrogel system is to identify fast gelation, or an accelerated photopolymerization (APP), based on the Bunsen Roscoe law (BRL) of reciprocity [35]. This reciprocity-law states that the effect of a photo-biological reaction is proportional only to the total irradiation dose, or fluence, (E_0_ = I_0_t), or the product of light intensity (I) and exposure time (t). To achieve the same efficacy, the required exposure time based on BRL is given by t = E_0_/I_0_, which gives the protocol for APP. For example, t = (30, 10, 5, 3, 2) min for I = (3, 9, 18, 30, 45) mW/cm^2^. The concept of APP has been clinically proven and commercialized in corneal crosslinking devices for many years [37]. However, there are no reported articles for a more general PPS. Validation of BRL has also been challenged by Lin’s non-linear law and the S-formulas for corneal crosslinking efficacy [36]. As described earlier, the steady-state efficacy of APP by higher light intensity has the drawback of lower efficacy than lower intensity. To overcome the drawback in AP, a PI concentration-controlled method (CCM) was proposed recently by Lin in corneal crosslinking [39], which may be extended for a more general PSS as follows. Gelation time may be shortened by raising the initial PI concentration or by increasing the light intensity, as also reported by Caludino et al. [19]. However, for smart hydrogel applications [2], the photoinitiator-induced cytotoxicity must be minimized, i.e., there is an optimal PI concentration for maximum efficacy and maximum cell viability.

### 4.5. Strategy for Improved Efficacy

We will discuss two classes of PPS: (a) synthetic polymer sample which is prepared in a hydrogel form, with PI pre-mixed in the gel uniformly; (b) nature polymer tissue (such as human cornea or other hydrogels), where the PI solution/drops may be administrated and diffused from its surface into the volume, having a controllable diffusion depth D. 

Class (a) is related to our system with a fixed F(z) = 1, and the initial PI concentration distribution can not be controlled by diffusion depth (D), for example, in photoinitiated polymerization of PEG-diacrylate hydrogel [4]. As shown by Figure 2 (right figure), the gelation spatial profile for the case of initially uniform PI concentration (with D ≫ 1.0 cm, or F = 1.0) is an increasing function of the depth (z), i.e., the anterior portion always has less efficacy (gelation). A strategy using focused light [29] and two-beam illumination [30] was proposed by Lin et al to improve the overall efficacy and uniformity of gelation. 

For class (b), the PI profile is controlled by the diffusion time (or the value of D). As shown by Figure 4 (left figure, with D = 0.5 cm) and Figure 3 (with D = 1.0 cm), both of which have a non-uniform PI, and a much more uniform crosslinked profile than that of a uniform PI, except that the center portion (about 0.5 to 1.2 cm), which has slightly high efficacy than both ends (near the surface and the bottom). Comparing Figure 3 (with C_0_ = 2.0 mM) and Figure 5 (with C_0_ = 3.0 mM), we found that higher C_0_ has a poor crosslinked spatial uniformity. However, low C_0_ also results in a low efficacy (comparing Figure 3; Figure 5). Therefore, an optimal C_0_ should range between 1.5 and 3.0 mM. 

Our Formula, Equation (5), predicts that faster type-I photoinitiated polymerization (gelation or crosslinking) may be achieved by using a higher intensity, which, however, also results in a low efficacy as shown in Figure 3. To overcome the drawback of low efficacy in accelerated process using a high intensity, as predicted by our S-formula, a CCM was proposed recently by Lin in corneal crosslinking [38]. Greater details based on the crosslink time (T*) and the S-formula, Equation (5), for a more general PSS, are discussed as follows.

As shown in Figure 4, higher light intensity has a faster rising efficacy, but a lower steady-state value due to its faster depletion of PI concentration. Therefore, a re-supply of PI drops, (with a supplying frequency defined as Fdrop), during the crosslink will improve the overall efficacy by a combined efficacy given by c – E_f_ = 1 − exp [−(S1 + S2 + S3 + …)], where Sj is the individual efficacy for each supply of PI. The time to re-supply PI is given by the the crosslink time (Tc) defined earlier which is inverse proportional to the light intensity, absorption coefficient (a), quantum yield (q), and the effective kinetic rate constant (g) for type-I pathway.

We note that the Fdrop is proposed by the combined consideration of the crosslink time (T_C_), defined by when the PI is highly depleted (or time needed to reach the steady-state of efficacy); and the surface S-value, which defines the strength of crosslink, or the number of re-applying PI drops needed to achieve the same S-value for all intensity ranges (5 to 100 mW/cm^2^). Lin’s proposed CCM [38] predicts the comparable efficacy (for the same dose) for intensity of 1.5 to 45 mW/cm^2^, based on a combined efficacy formula defined as c – E_f_ = 1−exp [−(S1 + S2 + … Sj)], with j = Fdrop, where Fdrop is given by Fdrop = (N − 1), with N given by [38], N = 0.816(I_0_/C_0_)^0.5^, based on a referenced light intensity, I_0_ = 3 mW/cm^2^ and C_0_ = 2 mM, that is N = 1.0 at the reference. For example, a combined steady-state efficacy with N = 3, or Fdrop = 2, for light intensity of 30 mW/cm^2^, where c – Ef f= exp[−(S1 + S2 + S3)], noting that the non-controlled efficacy is much lower than the CCM, curve (1 + 2 + 3). 

Compared to our previously published data (Figure 5 of Reference [37]), this study shows a similar profile, but under different parameters discussed as follows. We note that the steady-state efficacy spatial-profile is governed by C_0_F(z)exp(A_0_z), with A_0_ = 2.3a’C_0_ + Q. Therefore, similar profiles are expected when the PI initial concentration distribution, F(z) = 1 − 0.5/D, and A_0,_ are similar. In CXL (with thickness of 0.05 cm), stroma absorption (at UV 365 nm) Q = 32 cm^−1^, and a’ = 204 (%cm)^−1^, A_0_ = 79 cm^−1^, for C_0_ = 0.1%. In a thick polymer system having a thickness 20 times of cornea, or 1.0 cm, similar profiles as that of CXL require the conditions of: A_0_ is 20 times less than 79 cm^−1^, and z/D has the same ratio. For example, in a polymer system with Q = 0, we need a’ = 0.86 (mM cm)^−1^ and C_0_ = 2 mM; and D is 20 times of D (in CXL).

Our S-formula shows that, for the same PI concentration, the steady-state S for I_0_ = 20 mW/cm^2^ is 1.43 (or square-root of 2) lower than that of I_0_ = 10 mW/cm^2^, when no re-supply of PI is administered (or Fdrop = 0, N = 1). Under CCM, for example, in an APP having an intensity 30 mW/cm^2^, and C_0_ = 2 mM, we find N = 3, and Fdrop = N – 1 = 2, to achieve the same efficacy as the referenced intensity.

The major advantages and differences of our CCM compared to the conventional non-controlled method (NCM) high light intensity and PI concentration include: (i) in CCM, resupply of C_0_ is done at the crosslink time (T_C_) such that the masking effect due to surface layer extra C_0_ (which exists in NCM) is minimized; (ii) there is no waiting period in NCM during the light exposure, whereas our CCM allows a waiting period for enough PI diffusion, or large D value, which enhances the combined-efficacy; (ii) in NCM, too high an initial C_0_ achieves shallow crosslink depth, although it offers a high steady-state efficacy, as shown by Equation (5); whereas our CCM overcomes this drawback.

The proposed CCM for CXL has been partially clinically demonstrated [38]. However, it requires further measurements for general PPS, specially for thick polymers (1.0 to 1.5 cm). While our theoretical predictions provide quantitative guidance or strategy for optimal PPS, their accuracy depends on accurately measured parameters of absorption coefficient, quantum yield, and the effective kinetic rate constant for type-I pathway. Greater detail of CCM for specific polymer systems will be published elsewhere. 

As an extension of this study, the formulas developed and the coupled Equation (1) may be effectively used in other photopolymerization systems such as microfluidic device [23,27], Thiol-based photo-click collagen-PEG hydrogels [20,30] and to analyze the results of the prevention of oxygen inhibition of radical polymerization [42]. We note that the kinetic model and the formulas developed by Claudino et al. [20] were based on the assumption that both the photobase concentration and UV-light intensity are independent of polymer thickness, i.e., the conventional BLL. Therefore, their modeling is limited to optically-thin samples (<0.1 cm), and low PI concentration (<0.5 mM). The generalized BBL of this study will cover a much wider range of polymer thickness (0.1 to 1.5 cm) and PI concentration (0.2 to 5.0 mM). 

## 5. Conclusions

The overall PPS efficacy is proportional to the light dose (or fluence), the PI initial concentration and their diffusion depths. An optimal goal is to gain fast and maximum crosslink “volume”, or strength × depth, as well as polymerization uniformity. To summarize, this study presents several new findings including: non-BRL for crosslink depth, transient, and steady-state efficacy (or S-function), time-dependent generalized BLL and a new volume efficacy. For the first time, we have presented analytic formulas for curing depth and crosslink time without the assumption of thin film or spatial-average. Various optimal conditions are also developed for maximum efficacy based on a numerically-fit A-factor. Finally, the reported measurements of the role of light intensity and the PI concentration are well analyzed by our formulas.

## Figures and Tables

**Figure 1 polymers-11-00217-f001:**
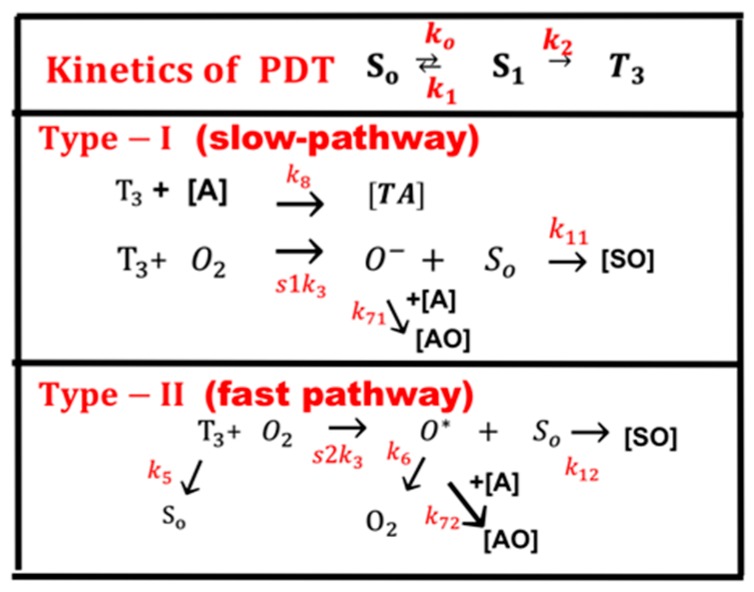
The kinetics of PDT, where [S_0_], [S_1_], and [T_3_] are the ground state, singlet excited state, and triplet excited state of PI molecules. Three pathways are shown for both the type-I and type-II processes. Ground state oxygen (O_2_) may couple to T_3_ to form either singlet oxygen (O*), or other reactive radical [O^−^]. In type-I pathway, T_3_ can interact directly with the collagen substrate (A); or with the oxygen (O_2_) to generate a superoxide anion (O^−^); in type-II pathway, T_3_ interacts with the ground oxygen (O_2_) to form a singlet oxygen (O*) [37].

**Figure 2 polymers-11-00217-f002:**
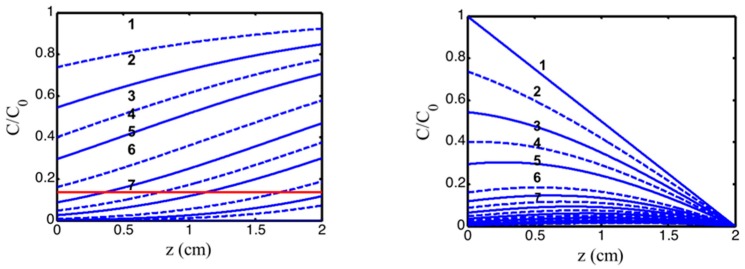
The normalized photoinitiator concentration profiles for uniform (**left figure**) and non-uniform (**right figure**, with D = 1.0 cm) PI initial distribution, for t = 0 (curves 1) and t = 50, 100, 150, 200, 300, 350 s (curves 2 to 7) with quantum yield q = 1.0, a’ = 0.2(mM·cm)^−1^, b’ = 0.15 (mM·cm)^−1^, and Q = 0.1 (1/cm) [34,35].

**Figure 3 polymers-11-00217-f003:**
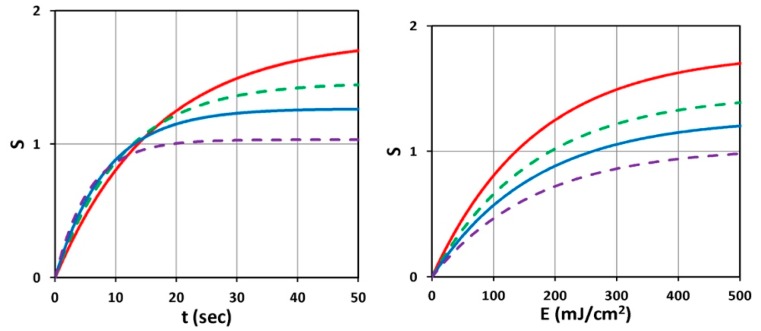
Time profiles of S versus t (**left figure**) and versus dose E_0_ (**right figure**) for various light intensity I_0_ = (10, 15, 20, 30) mW/cm^2^, shown by (red, blue, green, purple) curves, for D = 1.0 cm, C_0_ = 3 mM, at z = 0.5 cm.

**Figure 4 polymers-11-00217-f004:**
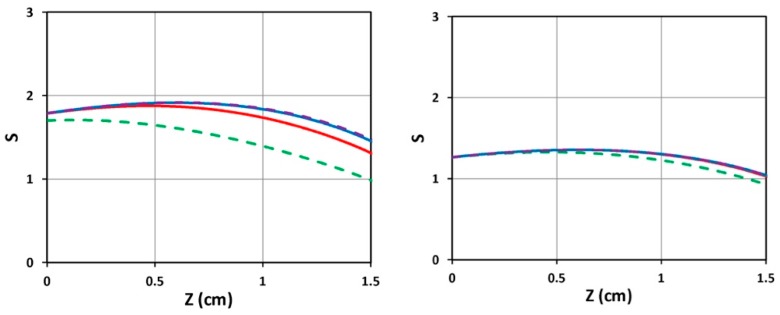
Spatial profiles of S for various exposure time t = (100, 200,3 00, 400) s shown by (green, red, blue, purple) curve, for light intensity I_0_ = 10 **(left figure**) and 20 (**right figure**) mW/cm^2^, and D = 1.0 cm, C_0_ = 2 mM.

**Figure 5 polymers-11-00217-f005:**
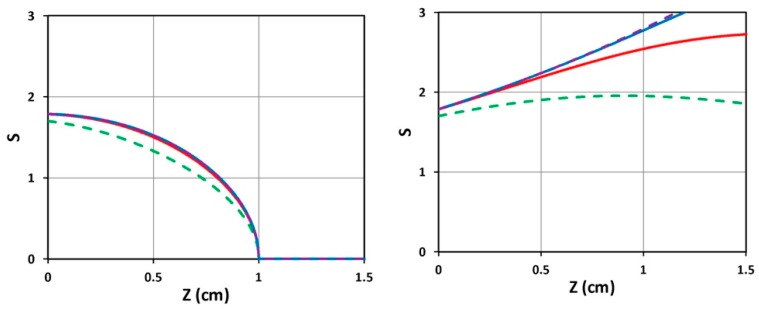
Spatial profiles of S for various exposure time t = (100, 200, 300, 400) s shown by (green, red, blue, purple) curve, for fixed light intensity I_0_ = 10 (mW/cm^2^) and C_0_ = 2 mM, for D = 1.0 cm (**left figure**) and D = 10 cm (**right figure**).

**Figure 6 polymers-11-00217-f006:**
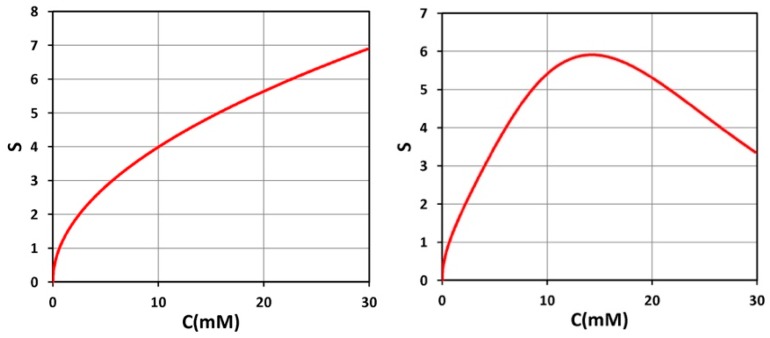
Steady-state S versus PI concentration (C_0_), for z = 0 (**left figure**) and z = 0.5 cm (**right figure**), for D = 1.0 cm, I_0_ = 10 mW/cm^2^.

**Figure 7 polymers-11-00217-f007:**
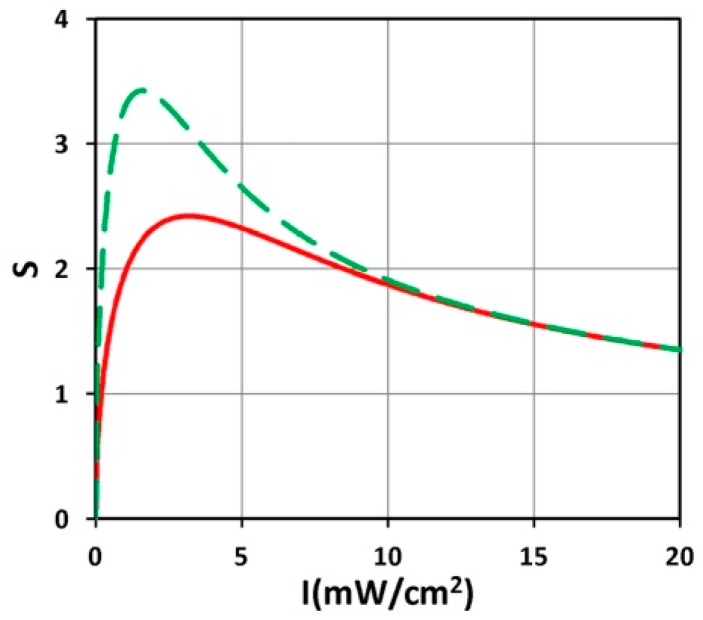
S versus light intensity (I_0_), for t = 100 s (red curve) and 200 s (green curve), and z = 0.5 cm, D = 1.0 cm, C_0_ = 2 mM.

**Figure 8 polymers-11-00217-f008:**
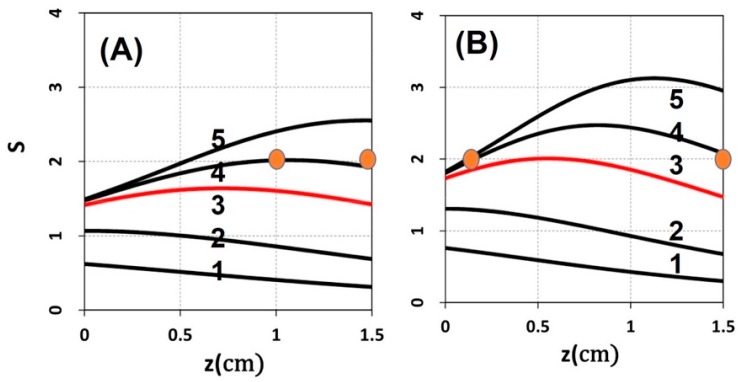
Spatial profiles of S for various PI concentration: (**A**) C_0_ = 2.0 mM, (**B**) C_0_ = 3.0 mM, at exposure time t = (30, 70, 165, 250, 400) s, with a’= 0.6, b’ = 0.1 (1/mM/cm), Q = 0.5 (1/cm), and I_0_ = 10 mW/cm^2^, using the analytic formula of Equation (5) with fit m = 0.001.

**Figure 9 polymers-11-00217-f009:**
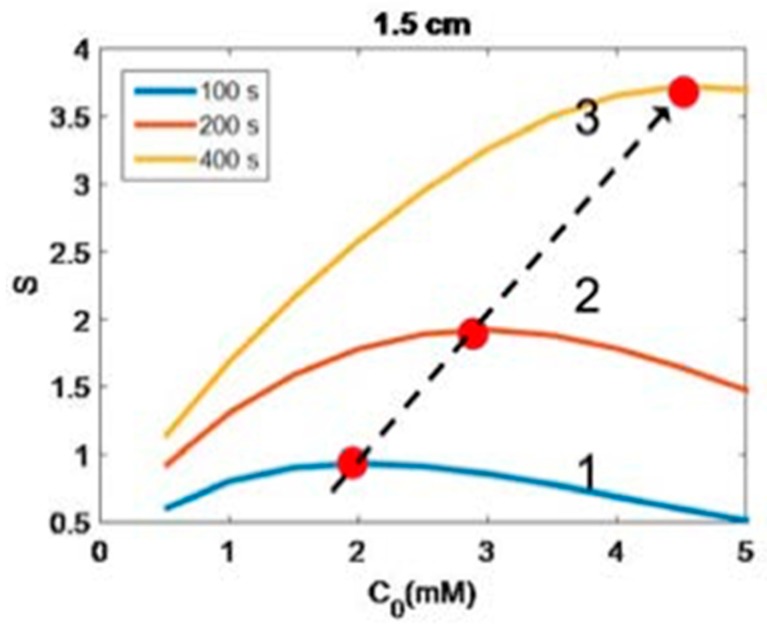
Numerical results of S versus PI concentration at various time, t = (100, 200, 400) s, for curves (1, 2, 3), at z = 1.5 cm; also shown are locations of the optimal concentrations (shown by red dots), which are increasing function of time (or light dose) (shown by dashed line).

**Table 1 polymers-11-00217-t001:** Abbreviations and formulas.

PPS	photoinitiated polymerization system
BRL	Bunsen–Roscoe reciprocal law
BLL	Beer–Lambert law
CXL	corneal collagen crosslink
PDT	photodynamic therapy
PI	photoinitiator
Generalized BLL for light intensity	I(z, t) = I_0_ exp[−(A_0_ – A_1_t)]z
	A_0_ = 2.3a’C_0_ + Q
	A_1_ = 2.3(a’ − b’)B’(1 − 0.5A’z)C_0_
	B’ = aqgI_0_
Type-I efficacy S-functions	S = [2KC_0_/B]^0.5^E’
	E’ = 1 − exp(−Bt)
	B = aqgI_0_ exp(−Az)
Crosslink depth (z_C_), first-order *	z_C_ = (1/A)ln(Y_0_)
	Y_0_ = KBtC_0_/d^2^
Crosslink Time (T_C_), first-order *	T_C_ = T_0_ exp(0.5Az)
	T_0_ = d/(aqgKC_0_I_0_)^0.5^

C_0_: initial PI concentration (at z = 0); E_0_ = tI_0_, light fluence (dose); I_0_ is initial surface (z = 0) light intensity; t is exposure time, q: quantum yield of PI triplet state; K is a rate constant; A: an averaged absorption factor, A = 1.15(a’ + b’)C_0_ + Q; Q: absorption of monomers (or substrate); d: threshold value for S-function. * Second-order formulas are shown in Equations (3) and (4).

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
