# Peer review of "Modeling the Optimal Conditions for Improved Efficacy and Crosslink Depth of Photo-Initiated Polymerization"

_polymers, 2019, doi:10.3390/polym11020217_

Round 1
Reviewer 1 Report
Title: Modeling the strategies for improved efficacy and crosslink depth of photo-initiated polymerization and phototherapy
The manuscript describes the theoretical approach of modeling the cross-linking process for optimization of the photo-polymerization. The optimization of photo-initiator makes possible minimize the amount of photo-initiator which is normally toxic material in biological-application, resulting in minimization of toxicity in the hydrogel.
In Addition, the manuscript describes separately the dynamics of photochemical reaction based on Type I and Type II photo-initiators and established the functions for each photo-polymerization process.
The contents of this manuscript described unique and potentially useful for some applications into the photo-polymerization process for biology. Thus, this manuscript has sufficient uniqueness in contents.
I recommend this manuscript has sufficient quality for publication.
Author Response
English was edited.
Reviewer 2 Report
The manuscript “Modeling the strategies for improved efficacy and crosslink depth of photo-initiated polymerization and phototherapy by J-T Lin et al. is a modified version of a previously submitted work. The authors have improved the manuscript, but the novelty of the work is still not clear and should be better clarified. In fact a figure similar to Fig. 7 was already reported in ref 35; it seems that the only difference is that in this work Fdrop=2 was set. The authors should explain what is the difference between the two models.
Moreover, other revisions are required before publication on Polymers.
- Line 39 (and line 356): It is not correct to use indiscriminately the terms ‘crosslinking’ and ‘click-chemistry’.
- Line 45: Please cite the work of J. Cabral et al. on the modeling of photopolymerization processes. Moreover, the model on the crosslinking time and depth (section 2.6 of the manuscript) should be compared with such work.
- Line 53: There are other important biomedical applications of photopolymerization, first of all the use of dental resins. Please correct this sentence.
- Line 60: The authors should specify and describe what is the corneal collagen crosslinking.
- Line 77: The authors state: “For optimal efficacy, strategy via controlled PI concentration will be presented, for the first time, where re-supply of PI concentration during the PPS is defined by a polymerization (crosslink) time which is inverse proportional to the light intensity.” However, the strategy of the re-supply of PI to increase the reaction efficacy was already reported by the authors (e.g., ref 35). The differences with published papers and the innovative aspects of this work should be presented and clarified.
- Many reference numbers are incorrect: please check all the reference list. Few examples are: ref 27, line 79, ref 26 line 106, ref 14 and 16 line 120 and 121, ref 26 line 173, ref 9, 13 and 29 line 302, ref 8 line 305, …
- Equation 1: Please describe all parameters used in the model. Many of them are not well defined and thus not clear for the reader.
- Line 119: The authors cite Fig. 1, but obviously they do not refer to Fig. 1 of this manuscript.
- Line 130: To which equation does ‘Eq. (c)’ refers to?
- Line 177: Please check the citing of Eq (3).
- Throughout the work, values are given to constant and parameters with no explanations; the authors should specify how such values are selected.
- The model presented should be confirmed by some experimental results.
Author Response
see fiel attached

Round 2
Reviewer 2 Report
The manuscript was improved after revision; however, some minor revisions are still required before publication on Polymers.
- Line 40: authors did not correct the indiscriminate use of the terms ‘crosslinking’ and ‘click-chemistry’ in this point.
- Please check that all acronyms are defined at their first use: for example, BLL is defined for a second and third time at line 99 and 171, BRL is not defined at line 206, CMM is defined for a second time at line 576, …
- Line 116: does ROS (in the text) correspond to SO (in Figure 1)?
- Line 125: which is the difference between [3O2] (line 125 and 127) and [O2] (line 139)?
- There are still many parameters/constants used that are not well defined and thus not clear for the reader. For example, what does F(z) correspond to (line 166)? What is M (line 229)?
- Line 185: it is better to specify that BLL can be considered as a special case of Lin-law for this specific application field.
- It is very confusing to cite equations and figures of published papers of the literature using the same nomenclature as in the manuscript, e.g., line 243, 246, 268-270, …
- Line 385 and 398: please complete the caption of Figure 5 and 9, repeating the caption of previous figures.
- There are some typos throughout the manuscript.
